# Turmeric Extract (*Curcuma longa*) Mediates Anti-Oxidative Effects by Reduction of Nitric Oxide, iNOS Protein-, and mRNA-Synthesis in BV2 Microglial Cells

**DOI:** 10.3390/molecules27030784

**Published:** 2022-01-25

**Authors:** Jana Streyczek, Matthias Apweiler, Lu Sun, Bernd L. Fiebich

**Affiliations:** 1Neuroimmunology and Neurochemistry Research Group, Department of Psychiatry and Psychotherapy, Medical Center, University of Freiburg, D-79104 Freiburg, Germany; jana.streyczek@uniklinik-freiburg.de (J.S.); matthias.apweiler@uniklinik-freiburg.de (M.A.); lu.sun@uniklinik-freiburg.de (L.S.); 2Faculty of Medicine, University of Freiburg, D-79104 Freiburg, Germany

**Keywords:** turmeric, curcuma, curcumin, NO, iNOS, oxidative stress, BV2, microglia

## Abstract

Plant-derived products have been used since the beginnings of human history to treat various pathological conditions. Practical experience as well as a growing body of research suggests the benefits of the use of turmeric (*Curcuma longa*) and some of its active components in the reduction of oxidative stress, a mechanism leading to neurodegeneration. In this current study, we investigated the effects of a preparation of *Curcuma longa*, and its constituents curcumin, tetrahydrocurcumin, and curcumenol, in one of the molecular pathways leading to oxidative stress, which is the release of NO, a free radical involved in stress conditions, using the BV2 microglial cell line. The concentration-dependent reduction of NO is linked to reduced amounts of iNOS protein- and mRNA-synthesis and is possibly mediated by the phosphorylation of mitogen-activated protein kinases (MAPK) such as p42/44 or p38 MAPK. Therefore, the use of turmeric extract is a promising therapeutic option for diseases linked to the dysregulation of oxidative stress, with fewer side-effects in comparison to the currently used pharmacotherapeutics.

## 1. Introduction

Turmeric, the dried yellow powder extracted from the rhizomes of *Curcuma longa*, has been used as a spice since ancient times, especially in Asian countries [1]. Moreover, it has gained importance due to its use in Ayurveda and in traditional Chinese medicine, as a phyto-therapeutic for the treatment of various conditions such as gastric and hepatic disorders, wound healing, and infectious diseases [2]. Practical experience as well as an emerging number of studies also implicate the involvement of turmeric and its active constituents in the regulation of oxidative stress and neuroinflammation [3]. On the molecular level, both mechanisms are associated with neurodegeneration [4,5], which can lead to different neurological and neuropsychiatric disorders such as Alzheimer’s disease, Parkinson’s disease, schizophrenia, and depression [6,7].

Under physiological conditions, cells of the central nervous system (CNS) are metabolically active, ensuring homeostasis and therefore showing an immense demand for energy. Since most of the required energy is provided by aerobic metabolism, levels of Reactive Oxygen Species (ROS) are increased within neuronal cells [8]. Effective systems for the elimination of ROS are antioxidants (e.g., vitamin A, C, and E), antioxidant minerals (e.g., zinc, selenium), and other free radical scavenging enzymes (e.g., SOD1, SOD2) [9], protecting cell constituents such as nucleic acids, proteins, or lipids from cell-threatening oxidative damage [10]. Oxidative stress occurs due to an evolving imbalance between the production of ROS and the described intracellular anti-oxidative defense mechanisms [9].

Nitric oxygen (NO) is a short-lived free radical highly involved in mechanisms regarding host defense and inflammation [11,12]. In response to stress conditions such as induction by lipopolysaccharides (LPS), inflammatory cytokines, or viral RNAs [13], members of the nitric oxide synthase (NOS) family produce large amounts of nitric oxygen out of l-arginine [14]. Out of the three existing isoforms, the inducible nitric oxide synthase (NOS2 or iNOS) is the predominant one in microglial cells. In contrast to the constitutively expressed family members, iNOS cannot be activated by intracellular calcium release or calmodulin, but is immunologically inducible [15]. Under physiological concentrations, NO constitutes an effective defense mechanism through killing bacteria, tumor cells, and viruses. Diversely, enormous release and intracellular accumulation of NO leads to damage of cells’ organelles and constituents [16,17]. Hence, the regulation of iNOS plays a pivotal role in the maintenance of homeostasis in the CNS, and its regulatory or cytotoxic functions are largely determined by the extent and duration of NO synthesis and the capacity of its elimination systems [11].

Microglial cells are considered as the immunomodulatory cells of the CNS. Responding to various pathological situations by phagocytosis and tissue repair, they ensure the protection of the brain’s parenchyma [18]. Contrarily, they also play a major role in neuroinflammatory and neurodegenerative diseases by initiating immune responses with harmful outcomes for the CNS [19]. Studies have shown that a variety of microglial functions are associated with different metabolic pathways induced by various key regulators, leading to different immune responses. Improvements in the knowledge of specific mediators and pathways in microglial activation can therefore lead to new treatment options regarding neurodegenerative diseases [19].

Many pathways have been shown to be involved in the regulation of iNOS expression. The phosphorylation of mitogen-activated protein kinases (MAPK) such as p42/44 MAPK (Erk 1/2) or p38 MAPK is highly postulated, proceeding in the activation of nuclear factor kappa-B (NF-κB). By binding to regulatory sides of the DNA and alternating its transcription, it is considered to be a key pro-inflammatory mediator [20,21]. 

Turmeric consists of a variety of active metabolites, including curcumin, tetrahydrocurcumin (THC), and curcumenol, amongst others. Even though many studies have shown that these derivates exert a wide range of anti-inflammatory, anti-proliferative, and anti-oxidative properties [22], their individual contribution to the observed effects is still unclear [23]. For instance, treatment of cadmium-induced neuroinflammation with an essential oil from turmeric rhizomes prevented the increase of pro-inflammatory cytokines such as IL-6 or TNFα and prevented a decrease of IL-10 as anti-inflammatory mediator [24]. Neuroprotective mechanisms were also exerted by the inhibition of acetylcholinesterase (AChE) activity after treatment with turmeric oils in the hippocampus and pre-frontal cortex; this treatment embraced the anti-inflammatory potential of acetylcholine [24,25]. Oily turmeric extract also showed protection against experimentally induced cere-bral ischemia by suppression of intracellular calcium levels. Additionally, ROS production as well as protein levels of the pro-apoptotic enzyme Bcl-2 were reduced [26], to name a few. 

Overall, turmeric interferes with many pathomechanisms involved in neuroinflammation, oxidative stress, and apoptosis, all leading to neurodegenerative diseases. It therefore displays high potential as a phyto-therapeutic, and thus further research on its targets and mechanisms as well as the role of its constituents is necessary. In the present study, we studied the effects of a turmeric extract (TE) on the regulation of oxidative stress in the BV2 microglial cell line by examining its effects on NO-release as well as iNOS protein synthesis and mRNA expression. Furthermore, we evaluated the effects of the active metabolites curcumin, THC, and curcumenol on NO-release to identify the constituent with a crucial impact on NO levels.

## 2. Results

### 2.1. Effects of TE and Its Constituents on Cell Viability

Primarily, we evaluated cell viability of BV2 microglial cells after treatment with different concentrations of TE with or without LPS-stimulation. Since TE is known to interfere with colorimetric assays based on the reaction of tetrazolium salt to formazan, we performed an ATP Luciferase-Assay to determine cell viability. As shown in Figure 1, we found a slight reduction in ATP content of the cells at the highest concentration of TE (500 µg/mL). Since no significant reductions of cell viability were observed, we used doses of up to 500 µg/mL, a dose commonly used in studies with turmeric.

Furthermore, we screened effects on cell viability after treatment with the curcuminoids curcumin (10 µM), THC (25 µM), and curcumenol (20 µM) by performing the MTT-Assay (data not shown) to measure the impact of the highest non-toxic concentrations on NO-decrease. The used concentrations of the curcuminoids did not significantly affect cell viability in the concentrations used. 

### 2.2. Effects of TE and Its Constituents on NO-Release

Next, we evaluated the release of NO in LPS-stimulated BV2 cells after treatment with different concentrations of TE and its constituents curcumin, THC, and curcumenol. Figure 2A shows the potent LPS-induced induction of NO-release in microglial cells, which was significantly and concentration-dependently reduced by treatment with TE in the concentrations of 100 µg/mL, 175 µg/mL, 250 µg/mL, and 500 µg/mL. Screening for non-toxic concentrations of curcumin (10 µM), THC (25 µM), and curcumenol (20 µM) showed a strong reduction of NO-production most effectively for curcumin followed by THC. No significant inhibitory effects were observed for curcumenol.

We therefore evaluated concentration-dependent effects of curcumin in LPS-stimulated BV2 cells (Figure 2B). Curcumin decreased LPS-induced NO-release with a significant and dose-dependent reduction using 5 µM and 10 µM curcumin. 

### 2.3. Effects of TE on iNOS-Protein Synthesis

We further evaluated the effects of TE on iNOS protein levels by Western Blot. LPS strongly induced iNOS protein synthesis in comparison to untreated cells, which was effectively and concentration-dependently reduced by the treatment with TE (Figure 3).

### 2.4. Effects of TE on iNOS-mRNA Expression

We next investigated if the inhibitory effects of TE on NO-release and iNOS-protein levels were mediated by the reduction of mRNA-levels of iNOS using qPCR. As shown in Figure 4, LPS potently induced iNOS mRNA expression in BV2 microglial cells compared to untreated cells. TE showed a strong dose-dependent decrease of LPS-induced iNOS mRNA levels when referenced to the housekeeping gene GAPDH. The reduction is significant for the three highest concentrations of 100 µg/mL, 250 µg/mL, and 500 µg/mL of TE.

### 2.5. Effects of TE on MAPK Activation

We further investigated signaling cascades associated with NO/iNOS-regulation such as phosphorylation of mitogen-activated protein kinases (MAPK, e.g., p38 MAPK, Erk 1/2). As shown in Figure 5A, phospho-p38 MAPK was reduced after treatment with all concentrations of TE, whereas phospho-Erk 1/2 (Figure 5B) was concentration-dependently decreased by TE, both with significant reduction of phosphorylation and thus activation at the highest concentration of 500 µg/mL TE.

## 3. Discussion

Turmeric has been used as a phyto-therapeutic for the treatment of many different symptoms and diseases for over 2000 years. Even though many effects on oxidative stress and neurodegeneration have been described, the underlying mechanisms and pathways as well as the impact of the different components are still only partially understood. Therefore, we are interested in the impact of TE and its main components curcumin, THC, and curcumenol on mechanisms leading to neurodegeneration. In the current study, we show that pretreatment with TE leads to a decrease of NO-release in LPS-stimulated BV2 cells that is mainly executed by the main component curcumin. Aligning with these results, the iNOS protein synthesis and mRNA-levels are significantly reduced. 

Initially, we evaluated the impact of TE on cell viability. The standard and established MTT assay used in our laboratory to determine cell viability did not provide usable results measuring cell toxicity effects of TE. This was due to the interference of TE with the reaction of tetrazolium salt to formazan, the underlying reaction of the assay. The same was observed in a then-performed LDH assay, a colorimetric assay, which is likewise based on a redox reaction but measured in the supernatant and not intracellularly. For curcumin, THC, and curcumenol, however, the MTT-assay reaction was not affected by the compounds.

The ATP assay, which is a chemiluminescent assay based on the firefly luciferase enzymatic reaction of Luciferin to Oxyluciferin using ATP, showed only slightly reduced amounts of ATP after applying 500 µg/mL of TE. LPS tends to upregulate the synthesis of ATP due to the stimulation of many different processes such as phosphorylation. Therefore, the results can be interpreted by high demands in energy as well as the enhancement of energy demands by applying TE. Since no toxic effects were observed, we used doses up to 500 µg/mL, a dose commonly used for studies with turmeric [1]. Additionally, it is worth mentioning that 60% of TE used in the current studies consists of the carrier substance maltodextrin, meaning that even smaller amounts of pure turmeric extract were used in the experiments.

TE significantly and concentration-dependently reduced NO-release in LPS-stimulated BV2 cells, with effects starting at the concentration of 100 µg/mL. Turmeric is known to consist of numerous curcuminoids, which are bioactive phenolic compounds. Out of the over 100 individual curcuminoids known by now, curcumin is the major component in turmeric and accounts for approximately 77% of the whole extract [23]. Therefore, it is not surprising that treatment of microglial cells with curcumin showed a strong reduction of NO-release starting at concentrations of 5 µM, equaling 3,7 µg/mL at the highest concentration of 10 µM, while 500 µg/mL of the used TE contained 145 µg/mL curcumin. Since the extract was bound to maltodextrin as a carrier substance counting for 60% of the final extract, the estimated effects of the total preparation were approximately 40 times higher than its component itself. Since curcumin is the main constituent of TE, its high potential regarding the decrease of NO-levels was an expected outcome. 

After testing cell viability for two other curcuminoids, THC, and curcumenol (data not shown), we screened their impact on the NO-release in the highest non-toxic concentration as well. We found a weak effect by THC and no effect by curcumenol, and thus these constituents most likely do not contribute to the NO-inhibiting potency of TE.

Studies have shown the inhibitory potency of curcuminoids in LPS-induced microglial cells on oxidative stress before. Curcumin, demethoxycurcumin, and bisdemethoxycurcumin all suppressed NO-production as well as iNOS mRNA in activated rat primary microglial cells, showing their important role in ensuring homeostasis and cell integrity [27]. Furthermore, peritoneal macrophages were found to produce high amounts of NO after stimulation with LPS, an effect that was reversed after treatment with TE [28]. 

It has been known for many years that NO is involved in several regulatory processes instead of solely functioning as a cell disruptive toxin [29]. However, its effects regarding cell protection and cell toxicity are still a topic of research and much discussed. One explanation for the controversial results is the role of NO, depending on its intra- and extracellular levels: physiological amounts exert neuroprotective effects, whereas exceeded concentrations result in neurotoxicity [30]. Both conditions lead to the activation or deactivation of different pathways and mediators. Additionally, various tissues are affected differently by NO release, supposing a tissue-dependent role of this molecule [11,29]. The described observations lead to a differentiated understanding of the role of NO in the pathomechanisms of various diseases induced by oxidative stress, along with neuroinflammation, leading to progressive neurodegeneration. The results might be interesting in research and treatment of psychiatric and neurological diseases such as Alzheimer’s disease (AD), Parkinson’s disease (PD), and depression, all linked to oxidative stress [31,32,33]. An extract from the rhizome of *Curcuma xantorrhiza Roxb* has been shown to exert anti-oxidative as well as anti-inflammatory effects [34], both considered hallmark pathologies of AD. Supporting these results, a standardized turmeric extract reduced the aggregation of β-Amyloid and Phosphorylated Tau Protein in in vitro ‘Alzheimer’ mice, suggesting an important role in the inhibition of accumulating plaques [35].

Pretreatment with TE significantly reduced protein synthesis of iNOS measured by Western Blot as well as mRNA-levels measured by qPCR in LPS-stimulated BV2 microglial cells. Reduction of iNOS-transcription with consecutive reduced protein synthesis leads to the observed reduced levels of NO. 

On a molecular level, turmeric has been known to modulate various cell-signaling pathways and to affect multiple targets [1,36]. Here, we focused on the regulation of protein kinases such as phosphorylated p38 mitogen-activated protein kinase (p38 MAPK) and phosphorylated p42/44 extracellular signal-dependent kinase (p42/44 = Erk 1/2). For both phosphorylated enzymes, we observed a significant reduction, especially after treatment with the highest concentration of 500 µg/mL of TE. The activities of turmeric on other pathways leading to neurodegenerative processes have been described before [37,38]. For instance, C-Jun-N-terminal kinases 1 and 2 (JNK1/2), other mitogen-activated protein kinases, show slightly enhanced phosphorylation after treatment with an essential oil obtained from *Curcuma zedoaria*. The same preparation leads to the inhibition of Akt/NF-κB signaling pathways in non-small cell lung carcinoma cell line [38]. Curcumin leads to the activation of NO–cGMP–K_ATP_ pathway in female Wistar rats, resulting in gastric protection [39]. With many pathways affected by turmeric and its active constituents, the main cascade leading to the synthesis of NO remains unclear and needs further research to be elucidated.

In conclusion, the importance of NO as a modulator of oxidative stress and neuroinflammation, both contributing to neurodegeneration, has grown immensely. Understanding the physiology and pathology of NO and the role of turmeric in the reduction of NO-levels, as well as the underlying pathways, might be key to treating diseases such as AD, PD, multiple sclerosis, or amyotrophic lateral sclerosis [40]. Therefore, future research is necessary to evaluate the potential of turmeric, its active constituents, and potential synthetic derivatives, as phyto-therapeutics, for instance, by using primary cells or organotypic hippocampal slices cultures (OHSC), which include both microglial and neuronal cells as well as in vivo disease models of neurodegeneration and neuroinflammation. 

## 4. Materials and Methods

### 4.1. Chemicals

The extraction of turmeric (TE obtained by Symrise AG, Holzminden, Germany) was carried out with hexane, and the crude extract was then adjusted to a specified color value of 27% using propylene glycol. To enable further use, the extract was bound to maltodextrin as a carrier substance (60% of the final extract). For our study, the extract was dissolved in distilled H_2_O and used in final concentrations of 50–500 µg/mL. Curcumin (10 µM), THC (25 µM), and curcumenol (20 µM) (all Cayman Chemical Company, Ann Arbor, MI, USA, distributed by BioMol, Hamburg, Germany) were dissolved and diluted into their final concentrations using DMSO (Merck KGaA, Darmstadt, Germany). Lipopolysaccharide (LPS) from E. coli (O127:B8; Sigma-Aldrich GmbH, Taufkirchen, Germany) was diluted in phosphate buffered saline (PBS; Roche Diagnostics, Mannheim, Germany) and used in a final concentration of 10 ng/mL in the cultures.

### 4.2. Microglia Cell Culture

BV2 microglial cells (kindly provided by Prof. Langmann, Department of Ophthalmology, University of Cologne, Cologne, Germany) were cultured in 1x RPMI 1640 Medium containing 5% fetal calf serum (FCS, Bio and SELL GmbH, Feucht/Nürnberg, Germany), 2 mM L-glutamine, and 1% penicillin/streptomycin (all cell culture solutions obtained by Gibco, Thermo Fisher Scientific, Bonn, Germany). The flasks were incubated in a humidified atmosphere at 37 °C with 5% CO_2_. The cells were routinely passaged by trypsinization and re-seeded into 6-, 12-, 24-, or 96-well plates. On the following day, medium was replaced by fresh medium, and the plates were used for further experimental treatment. 

### 4.3. Determination of NO-Release from LPS-Activated BV2 Cells

BV2 cells were pretreated with TE (50 to 500 µg/mL) and its constituents curcumin (10 µM), THC (25 µM), and curcumenol (20 µM) for 30 min, respectively. Then, cells were stimulated with 10 ng/mL LPS and incubated for 24 h. Supernatants were harvested, and NO-levels were measured using the commercially available Griess Reagent System (Promega Corporation, Fitchburg, MA, USA) following the manufacturer’s protocol. Briefly, a NO standard curve was prepared and standards as well as supernatants of the samples were transferred to a 96-well microplate. Sulfanilamide solution was added to all wells, and the plate was incubated for 10 min in the dark. Then, N-1-naphthylethylenediamine dihydrochloride (NED) solution was added to all wells and the plate was incubated for 10 min again. The plate was read at 530 nm using a colorimetric microplate reader (MRX^e^ Microplate reader, Dynex Technologies, Denkerdorf, Germany).

### 4.4. Cell Viability Assays

Cell viability in BV2 microglial cells after treatment with different concentrations of TE (50 to 500 µg/mL), curcumin, THC, and curcumenol was measured using a colorimetric MTT assay (Sigma-Aldrich GmbH, Taufkirchen, Germany). Generally, this assay determines the number of metabolically active and viable cells in cell culture based on the reduction of a yellow tetrazolium salt (3-(4,5-dimethylthiazol-2-yl)-2,5-diphenyltetrazolium bromide or MTT) to purple formazan in the cells. The procedure corresponded to experiments already performed and described [41,42].

Due to the inconclusive results, we additionally performed an ATP assay (CellTiter-Glo^®^ 2.0 Assay; Promega Corporation, Fitchburg, MA, USA) as a further method to measure cell viability after treatment with TE following the manufacturer’s protocol. Since ATP is produced by living cells, its concentration is a widely accepted indicator for metabolically active and therefore viable cells. The chemiluminescent assay is based on the firefly luciferase enzymatic reaction of Luciferin to Oxyluciferin using ATP while producing luminescent light, which can be measured using a luminometer (Modulus^TM^ Microplate Multimode Reader; Promega Corporation, Fitchburg, MA, USA).

### 4.5. RNA Isolation and Quantitative PCR

For quantification of iNOS mRNA expression, quantitative real-time PCR (qPCR) was performed in BV2 cells. Cultured cells were pre-treated with TE (50–500 µg/mL) for 30 min and then stimulated with LPS (10 ng/mL) for 4 h. RNA was extracted according to the manufacturer’s protocol using the GeneMATRIX Universal RNA Purification Kit (Roboklon GmbH, Berlin, Deutschland). Afterwards, cDNA was reverse-transcribed from 500 ng of total RNA in a 30 μL total reaction volume with initial denaturation at 70 °C (10 min) with a following amplification cycle after addition of 10 µL master mix. qPCR amplification was calculated by the CFX96 real-time PCR detection system (Bio-Rad Laboratories GmbH, Feldkirchen, Germany). The housekeeping gene glyceraldehyde 3-phosphate dehydrogenase (GAPDH) served as an internal control for sample normalization. The primer sequences were GAPDH: Forward (Fw): 5′-TGGGAAGCTGGTCATCAAC-3′/Reverse (Rv): 5′-GCATCACCCCATTTGATGTT-3′, and iNOS Fw: 5′-GAGCCCACCGGACATG-3′/Rv: 5′-CAGATGGTGGCCTCCC-3′. Primers were designed using Universal ProbeLibrary Assay Design Center (Roche Diagnostics, Mannheim, Germany) and obtained by biomers.net GmbH (Ulm, Germany).

### 4.6. Immunoblotting

BV2 microglial cells were pre-treated with different concentrations of TE for 30 min followed by stimulation with 10 ng/mL LPS for another 30 min examining MAPK phosphorylation or for 24 h investigating iNOS protein levels. Afterwards, cells were washed with cold PBS and lysed mechanically in lysis buffer (42 mM Tris–HCl, 1.3% sodium dodecyl sulfate, 6.5% glycerin, 100 μM sodium orthovanadate, 2% phosphatase and 0.2% protease inhibitors). The measurement of protein concentrations was performed according to the manufacturer’s instructions using the bicinchoninic acid (BCA) protein assay kit (Thermo Fisher Scientific, Bonn, Germany). For Western Blotting, 20 μg of total protein from each sample was subjected to sodium dodecyl sulfate-polyacrylamide gel electrophoresis (SDS-PAGE) under reducing conditions. Afterwards, proteins were transferred onto polyvinylidene fluoride (PVDF) membranes (Merck Millipore, Darmstadt, Germany) by semi-dry blotting and next blocked with Roti-Block (Roth, Karlsruhe, Germany). Membranes were then incubated overnight covered in primary antibodies (rabbit NOS2 (sc-651, 1:500; Santa Cruz Biotechnology, Heidelberg, Germany); rabbit anti-p38 MAPK (9211S, 1:1000; Cell Signaling Technology, Frankfurt, Germany); rabbit anti-phospho-Erk 1/2 (9101S, 1:1000; Cell Signaling Technology, Frankfurt, Germany); mouse anti-β-actin (sc-47778, 1:5000; Santa Cruz Biotechnology, Heidelberg, Germany)). On the following day, the proteins were detected with horseradish-peroxidase-coupled goat anti-rabbit IgG or sheep anti-mouse IgG (1:25,000/1:20,000 dilution; both Amersham, GE Healthcare, Freiburg, Germany) using enhanced chemiluminescence (ECL) reagents (Biozym, Hessisch Oldendorf, Germany). Densitometric analysis was performed using ImageJ software (National Institute of Health, Bethesda, MD, USA). 

### 4.7. Statistical Analysis

Raw values were converted to percentage of LPS (10 ng/mL)-treated cells or the appropriate positive control, such as untreated cells in the MTT- and ATP-assay. Data are represented as mean ± standard error of the mean (SEM) of at least three independent experiments. The statistical comparisons were performed using one-way ANOVA with Bonferroni post hoc test (Prism 8 software, GraphPad software Inc., San Diego, CA, USA). The level of significance was set at * *p* < 0.05, ** *p* < 0.01, *** *p* < 0.001 and **** *p* < 0.0001.

## 5. Conclusions

Turmeric extract (TE) exerts strong anti-oxidative effects in BV2 microglial cells by significantly and concentration-dependently reducing LPS-induced NO-release. The effect is mainly mediated by its main constituent curcumin, but other so far not identified components might contribute as well. Furthermore, TE strongly reduced iNOS mRNA expression and iNOS protein synthesis, probably due to reduced phosphorylation and thus activation of mitogen-activated protein kinases p42/44 MAPK (Erk 1/2) and p38 MAPK. Therefore, turmeric and its active constituents display high potential as phyto-therapeutics in the treatment of CNS diseases correlated to oxidative stress, such as Alzheimer’s disease, Parkinson’s disease, and schizophrenia. 

## Figures and Tables

**Figure 1 molecules-27-00784-f001:**
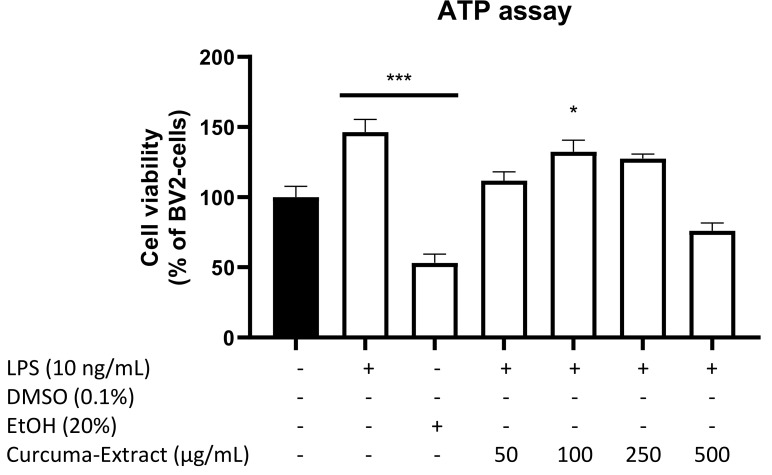
Effects of TE on cell viability of LPS-stimulated BV2 cells (ATP assay). TE was added 30 min before stimulation with 10 ng/mL LPS for 24 h. ATP levels and thus cell viability were measured by production of luminescent light following the firefly luciferase enzymatic reaction of Luciferin to Oxyluciferin using ATP. Values are presented as the mean ± SEM of four independent experiments. Statistical analysis was performed using one-way ANOVA with Bonferroni post hoc test and * *p* < 0.05 and *** *p* < 0.001, compared to untreated cells.

**Figure 2 molecules-27-00784-f002:**
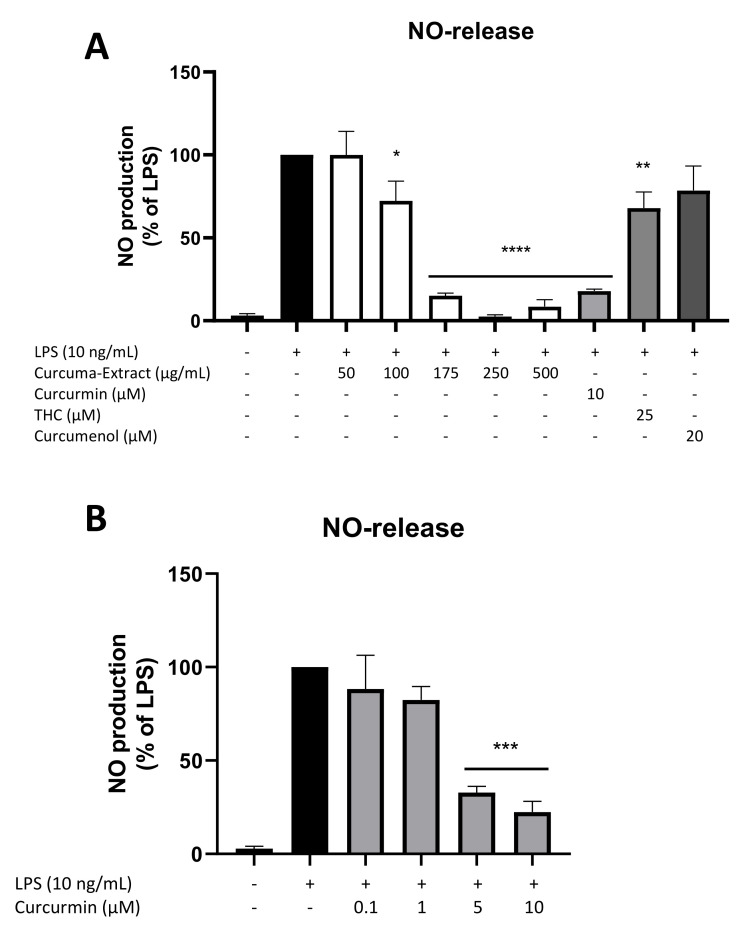
Effects of different concentrations of TE and its constituents curcumin, THC, and curcumenol (one single high dose, respectively) (**A**), and different doses of curcumin (**B**), on NO-release in LPS-stimulated BV2 cells. The substances were added 30 min before stimulation with 10 ng/mL LPS. NO-release was measured as described below. Values are presented as the mean ± SEM of at least three independent experiments. Statistical analysis was performed using one-way ANOVA with Bonferroni post hoc tests and * *p* < 0.05, ** *p* < 0.01, *** *p* < 0.001, and **** *p* < 0.0001 compared to LPS.

**Figure 3 molecules-27-00784-f003:**
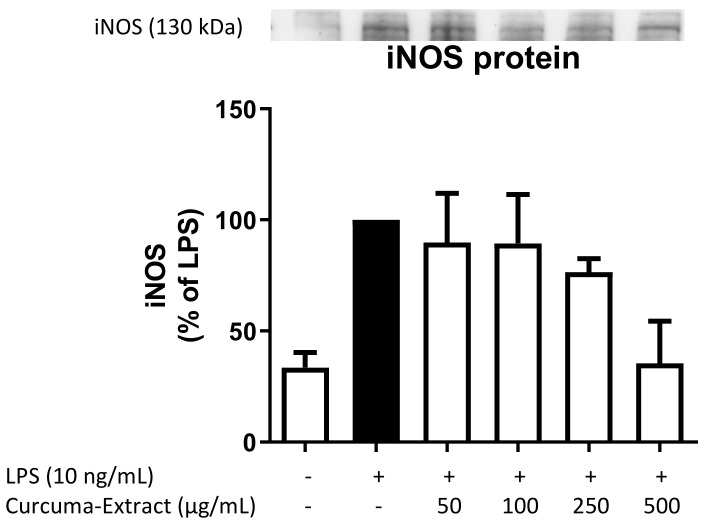
Effects of TE on iNOS protein levels in LPS-stimulated BV2 cells. TE was added 30 min before stimulation with 10 ng/mL LPS. After 24 h, Western Blot analysis was performed. Values are presented as the mean ± SEM of three independent experiments, and statistical analysis was performed using one-way ANOVA with Bonferroni post hoc test compared to LPS.

**Figure 4 molecules-27-00784-f004:**
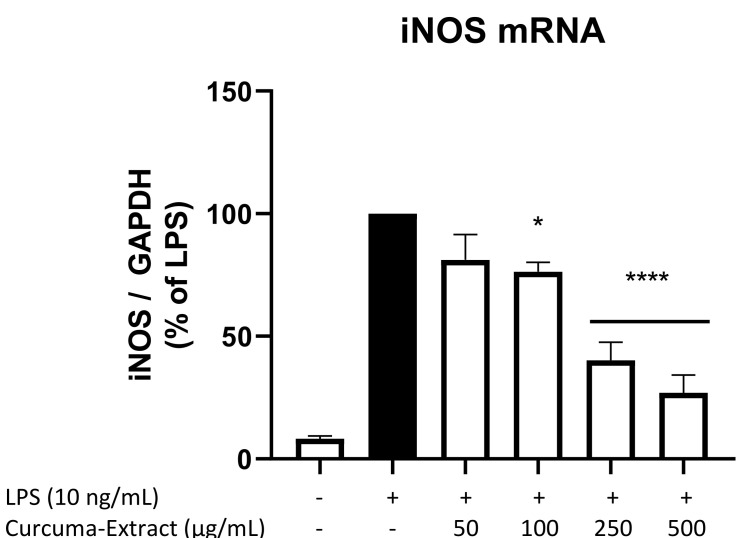
Effects of TE on iNOS mRNA expression in LPS-stimulated BV2 cells. TE was added 30 min before stimulation with 10 ng/mL LPS for 4 h. RNA was then isolated, and iNOS mRNA levels were determined using qPCR. Values are presented as the mean ± SEM of at least three independent experiments. Statistical analysis was performed using one-way ANOVA with Bonferroni post hoc tests and * *p* < 0.05 and **** *p* < 0.0001, compared to LPS.

**Figure 5 molecules-27-00784-f005:**
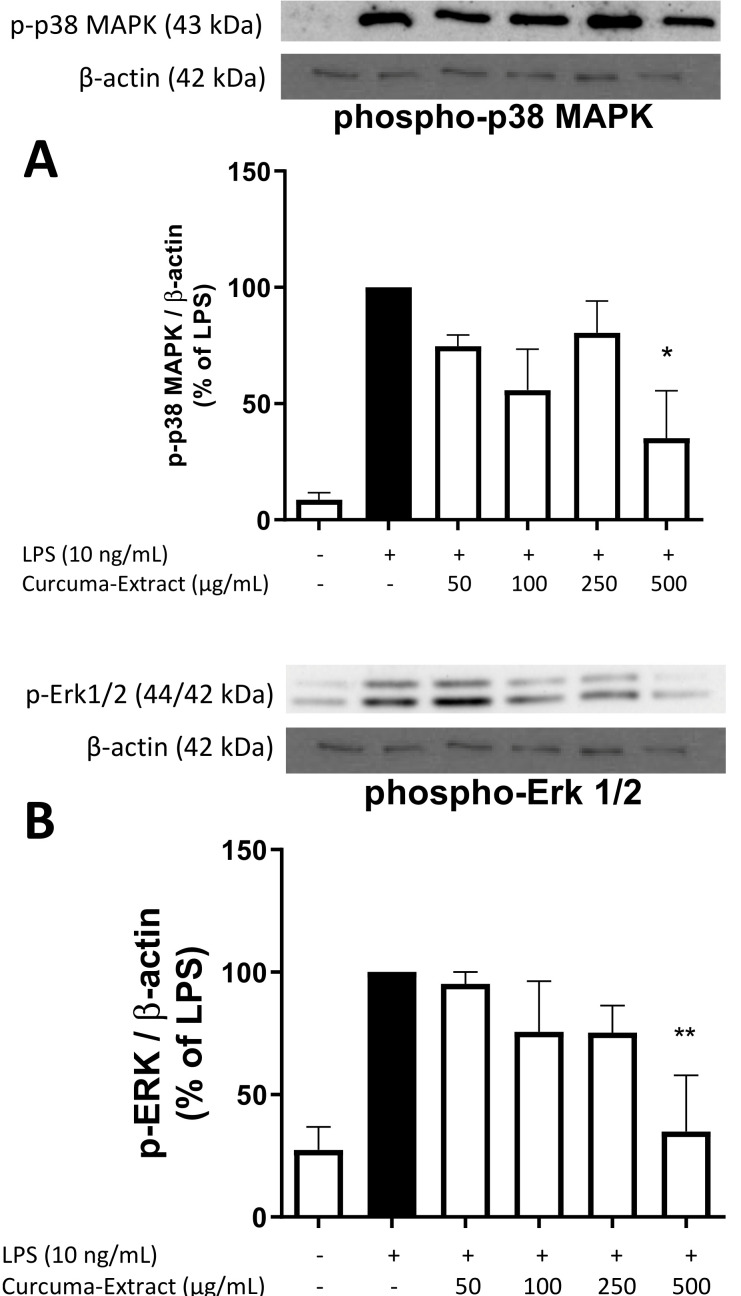
Effects of TE on p38 MAPK (**A**) and Erk 1/2 (**B**) phosphorylation in LPS-stimulated BV2 cells. TE was added 30 min before stimulation with 10 ng/mL LPS for 30 min. Western Blot analysis was performed for p-p38/MAPK (**A**) and phospho-Erk 1/2 (**B**). Values are presented as the mean ± SEM of at least three independent experiments, and protein levels were referenced to β-actin. Statistical analyses were performed using one-way ANOVA with Bonferroni post hoc test and * *p* < 0.05 and ** *p* < 0.01, compared to LPS.

## Data Availability

The data presented in this manuscript are available from the corresponding author upon request.

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
