# Peer review of "Turmeric Extract (Curcuma longa) Mediates Anti-Oxidative Effects by Reduction of Nitric Oxide, iNOS Protein-, and mRNA-Synthesis in BV2 Microglial Cells"

_molecules, 2022, doi:10.3390/molecules27030784_

Round 1
Reviewer 1 Report
The reviewer basically agrees with the publication of this article, but some questions need to be addressed:
- Please modify the article in strict accordance with the template of Molecules.
- The text in the Western blot can't be seen clearly.
- Please strictly keep the case of the first letter of the keyword consistent.
- What does the mean ± SEM indicate? mean ± SD?
Author Response
The reviewer basically agrees with the publication of this article, but some questions need to be addressed:
- Please modify the article in strict accordance with the template of Molecules.
- The text in the Western blot can't be seen clearly.
- Please strictly keep the case of the first letter of the keyword consistent.
- What does the mean ± SEM indicate? mean ± SD?
Answer: Dear Reviewer 1, many thanks for your positive feedback for our paper. Regarding the points you noted:
- We adjusted the manuscript in order to align with the template of Molecules.
- We have again revised the legends of the corresponding graphics to make them easier to read.
- All keywords are now written consistently with capital letters.
- Standard deviation (SD) and standard error of the mean (SEM) are two different statistical measures. The SD indicates the dispersion of individual measured values of a sample from its mean value, whereas the SEM classifies the precision of the mean value of a sample. It follows the formula, where s is the standard deviation of the sample and n is the sample size. Since we expect large samples and low dispersion of the measured values in their population, we have chosen to use the standard error of the mean to calculate our statistics in this paper. In addition, we have introduced the abbreviation standard error of the mean (SEM) in Section 7. Statistical analysis to clarify the difference with SD.
Reviewer 2 Report
The manuscript of Streyczek describes the effects of turmeric extracts on oxidative stress, measured by NOS release, in glial cells. Although the manuscript displays interesting evidence, however, it is not suitable for publication in this form and requires some revisions:
1) for all experimental points, the authors should measure the effects of TE (at least 500 µg/ml) without LPS treatment on cell vitality;
2) in figure 3, it is not clear how the authors have referenced the blots for the iNOS protein synthesis. The authors have to normalize the bands of iNOS versus the blots for actin or GPDH or other housekeeping genes as performed for the following p38 or p42/44 MAPK immunoblottings.
Author Response
The manuscript of Streyczek describes the effects of turmeric extracts on oxidative stress, measured by NOS release, in glial cells. Although the manuscript displays interesting evidence, however, it is not suitable for publication in this form and requires some revisions:
1) for all experimental points, the authors should measure the effects of TE (at least 500 μg/ml) without LPS treatment on cell vitality;
2) in figure 3, it is not clear how the authors have referenced the blots for the iNOS protein synthesis. The authors have to normalize the bands of iNOS versus the blots for actin or GPDH or other housekeeping genes as performed for the following p38 or p42/44 MAPK immunoblottings.
Answer: Dear Reviewer 2, thanks for your helpful comments regarding the improvements of our manuscript.
1. Induction with lipopolysaccharide (LPS) imposes a tremendous stress condition on cells and thereby induces a cellular response, such as an increased production of NO. The presence of an appropriate stress situation for the cells is a basic requirement for the development of neurodegenerative diseases, which we address in our paper. For this reason, we decided against showing the effect of TE on cell viability of microglial cells alone and to present it only in combination with LPS.
2. Before starting the Western blot, we perform a standard protein concentration determination in our laboratory using a colorimetric BCA protein assay. This accurate determination is essential for an intensity comparison of the bands displayed in the Western blot, which can only be used if equivalent amounts of protein are used. Therefore, in the context of this referencing already performed, we did not perform the additional referencing with actin.
Reviewer 3 Report
I agree with the Acceptance of the paper for publication, justified on the basis of:
- the subject of the research is very topical, and brings to the fore the importance of research in the field of medicinal plants, in the context of identifying new therapeutic values for which turmeric species can be associated in preventive phytotherapy;
- the work is easy for a specialist in the field, the results obtained are well justified and presented;
- the working methods used are adequate to the proposed objective, all stages of the research are detailed, modern methods of analysis in the acellular system are used, with the evaluation of the plant extract compartment and the dominant constituents in the modulation of oxidative erasure, as the main disruptive factor at the neuronal level;
- the behavior of turmeric extract and active constituents, curcumin, tetrahydrocurcumin, curcumenol, on molecular pathways responsible for inducing oxidative stress were evaluated (studies on the production of NO, iNOS, mRNA, MAPK, in cellular models - BV2 microglial cells) involved in neurodegeneration;
- the evaluation of the action of the extract and of the active chemical constituents on the oxidative stress aimed at establishing a direct causality between the concentration of active chemical constituents and the potential synergy effects with respect to the chemical constituents of the phytocomplex; all the results obtained are statistically processed;
- the diagrams used in the expression of the results are justifying, highlighting very well the dose-time relationship in the negative modulation of the stressors generating neurodegradation;
- the discussions are relevant and refer to each objective proposed in the research; each stage of the work is discussed in detail, the scientific observations being of great relevance for the study approached;
- the conclusions are in line with the aim of the study, Turmeric extract as the authors point out may be a therapeutic perspective in reducing neurodegradation, as a direct involvement of oxidative stress in various types of neurodegeneration, with side effects clearly inferior to synthetic drugs;
- the paper is supported by current data from the literature.
Author Response
I agree with the Acceptance of the paper for publication, justified on the basis of:
- the subject of the research is very topical, and brings to the fore the importance of research in the field of medicinal plants, in the context of identifying new therapeutic values for which turmeric species can be associated in preventive phytotherapy;
- the work is easy for a specialist in the field, the results obtained are well justified and presented;
- the working methods used are adequate to the proposed objective, all stages of the research are detailed, modern methods of analysis in the acellular system are used, with the evaluation of the plant extract compartment and the dominant constituents in the modulation of oxidative erasure, as the main disruptive factor at the neuronal level;
- the behavior of turmeric extract and active constituents, curcumin, tetrahydrocurcumin, curcumenol, on molecular pathways responsible for inducing oxidative stress were evaluated (studies on the production of NO, iNOS, mRNA, MAPK, in cellular models - BV2 microglial cells) involved in neurodegeneration;
- the evaluation of the action of the extract and of the active chemical constituents on the oxidative stress aimed at establishing a direct causality between the concentration of active chemical constituents and the potential synergy effects with respect to the chemical constituents of the phytocomplex; all the results obtained are statistically processed;
- the diagrams used in the expression of the results are justifying, highlighting very well the dose-time relationship in the negative modulation of the stressors generating neurodegradation;
- the discussions are relevant and refer to each objective proposed in the research; each stage of the work is discussed in detail, the scientific observations being of great relevance for the study approached;
- the conclusions are in line with the aim of the study, Turmeric extract as the authors point out may be a therapeutic perspective in reducing neurodegradation, as a direct involvement of oxidative stress in various types of neurodegeneration, with side effects clearly inferior to synthetic drugs;
- the paper is supported by current data from the literature.
Answer: Dear Reviewer 3, many thanks for the detailed review of our manuscript and the positive feedback.
Reviewer 4 Report
The manuscript focused on the anti-oxidative effect of Turmeric extract (Curcuma longa) and further discussed its effect on BV2 microglial cells. The viability of BV2 microglial cells is determined by performing an MTT assay. The data on the regulation of iNOS expression is well presented in Western Blot analysis. Furthermore, the iNOS-mRNA expression levels are determined using qPCR and are well documented in the present article. The authors have presented the article in a very nice way; the work done is to the mark of the journal.
The work done is up to the mark, therefore, I suggest acceptance of the article.
Author Response
The manuscript focused on the anti-oxidative effect of Turmeric extract (Curcuma longa) and further discussed its effect on BV2 microglial cells. The viability of BV2 microglial cells is determined by performing an MTT assay. The data on the regulation of iNOS expression is well presented in Western Blot analysis. Furthermore, the iNOS-mRNA expression levels are determined using qPCR and are well documented in the present article. The authors have presented the article in a very nice way; the work done is to the mark of the journal.
The work done is up to the mark, therefore, I suggest acceptance of the article.
Answer: Dear Reviewer 4, thank you for your positive feedback, we appreciate the time and consideration you took to review our manuscript.
Round 2
Reviewer 2 Report
No comments